# Prenylated quinolinecarboxylic acid compound-18 prevents sensory nerve fiber outgrowth through inhibition of the interleukin-31 pathway

Masato Ogura *, Kumiko Endo, Toshiyuki Suzuki, Yoshimi Homma

Fukushima Medical University School of Medicine, Fukushima, Japan

* masato57@fmu.ac.jp

## Abstract

Interleukin-31 (IL-31) is involved in excessive development of cutaneous sensory nerves in atopic dermatitis (AD), leading to severe pruritus. We previously reported that PQA-18, a prenylated quinolinecarboxylic acid (PQA) derivative, is an immunosuppressant with inhibition of p21-activated kinase 2 (PAK2) and improves skin lesions in Nc/Nga mice as an AD model. In the present study, we investigate the effect of PQA-18 on sensory nerves in lesional skin. PQA-18 alleviates cutaneous nerve fiber density in the skin of Nc/Nga mice. PQA-18 also inhibits IL-31-induced sensory nerve fiber outgrowth in dorsal root ganglion cultures. Signaling analysis reveals that PQA-18 suppresses phosphorylation of PAK2, Janus kinase 2, and signal transducer and activator of transcription 3 (STAT3), activated by IL-31 receptor (IL-31R), resulting in inhibition of neurite outgrowth in Neuro2A cells. Gene silencing analysis for PAK2 confirms the requirement for STAT3 phosphorylation and neurite outgrowth elicited by IL-31R activation. LC/MS/MS analysis reveals that PQA-18 prevents the formation of PAK2 activation complexes induced by IL-31R activation. These results suggest that PQA-18 inhibits the IL-31 pathway through suppressing PAK2 activity, which suppresses sensory nerve outgrowth. PQA-18 may be a valuable lead for the development of a novel drug for pruritus of AD.

## Introduction

Atopic dermatitis (AD) is known as chronic inflammatory dermatitis with severe pruritus [1–3]. Pruritus often causes scratching behavior, leading to disturbance of sleep and exacerbation of AD [4,5]. Since pruritus reduces the quality of life of AD patients, and the intensity of pruritus negatively correlates with psychosocial well-being, suppression of pruritus plays an important role in the improvement of AD [5,6]. However, it has been reported that most pruritus in AD is not suppressed by antihistamines [5,7], and the pruritus leads to the prolongation of AD treatment and the occurrence of side effects due to long-term administration [5,7,8]. Although the underlying mechanism of pruritus development in AD is not fully understood, recent

**Funding:** This work was funded by the Japan Society for the Promotion of Science (Grant-in-Aid for Scientific Research) [grant number 18K06702 (MO): https://kaken.nii.ac.jp/ja/index/] and Fukushima Medical University (grant for Project Research) [grant number KKI29001-1 (MO)]: https://www.fmu.ac.jp].

**Competing interests:** The authors have declared that no competing interests exist.

studies have demonstrated that excessive development of cutaneous sensory nerves and production of sensory nerve stimulants, such as interleukin-31 (IL-31), are involved in the development of pruritus [9–11]. Therefore, there is a demand for the development of new therapeutic agents for AD that target the cutaneous sensory nerves and IL-31.

IL-31 is a cytokine mainly produced by activated helper type 2 T cells, and the level of IL-31 is increased in lesional skin of patients with AD and Nc/Nga mice as an AD model [12–14]. IL-31 activates the Janus kinase (JAK)/signal transducer and activator of the transcription (STAT) pathway through the IL-31 receptor (IL-31R), which is composed the IL-31Rα subunit and oncostain M receptor [5]. IL-31R is reported to be highly expressed in sensory neurons, and activates sensory nerves, promotes the development of sensory nerve fibers, and lowers the pruritus threshold [9,15]. Indeed, in addition to Nc/Nga mice, IL-31 transgenic mice have been shown to develop increased scratching behavior, enhanced sensory nervous system and atopic-like dermatitis [15]. Moreover, intradermal administration of IL-31 also causes severe pruritus in normal mice [9,16]. Thus, suppressing the development of the sensory nerve density in AD and inhibiting the IL-31 pathway related to development could lead to attenuation of pruritus in AD.

Cellular slime molds are soil microorganisms that produce many pharmacologically active molecules and are an important source of lead compounds for medical research [17–22]. We have previously identified prenylated quinolinecarboxylic acid (PQA)-18, as a novel immunosuppressant with p21-activated kinase 2 (PAK2) inhibitory activity, from a group of slime mold-derived PQA derivative [21]. We reported that the application of PQA-18 ointment to lesional skin improved dermatitis and scratching behavior in Nc/Nga mice [21]. In this study, we examined the effect of PQA-18 on the cutaneous sensory nervous system in order to elucidate the mechanism of PQA-18 to improve dermatitis. PQA-18 suppressed the excessive development of sensory nerves in the lesional skin of Nc/Nga mice. Furthermore, PQA-18 suppressed IL-31-induced neurite outgrowth in dorsal root ganglion (DRG) neurons. Analysis of the effect of PQA-18 on the IL-31 pathway revealed that PQA-18 suppresses activation of PAK2, JAK2 and STAT3 induced by IL-31R stimulation. IL-31-induced neurite outgrowth and STAT3 activation were inhibited by suppressing PAK2 expression. These results suggest that PQA-18 suppresses excessive development of cutaneous sensory nerves through inhibition of the IL-31 pathway. PQA-18 may be a promising compound for improving pruritus of AD.

## Materials and methods

### Antibodies and chemicals

The PQA-18 in this study was synthesized and purified as previously described [20,21], and the structure and purity were confirmed by $^1$H and $^{13}$C NMR spectroscopy and high-resolution mass spectroscopy. The purity of PQA-18 was greater than 98%. Mouse anti-β-actin (A5316) monoclonal antibody (mAb), mouse anti-βIII-tubulin (05–559) mAb, and dimethyl sulfoxide (DMSO) were purchased from Sigma-Aldrich (St. Louis, MO); rabbit anti-phospho-STAT3 (Tyr705) (#9145) mAb, rabbit anti-STAT3 (#12640) mAb, rabbit anti-phospho-JAK2 (Tyr1008) (#8082) mAb, rabbit anti-JAK2 (#3230) mAb, rabbit anti-PAK2 (#2608) polyclonal antibody, rabbit anti-phospho-PAK2 (Ser141) (#2606) polyclonal antibody were obtained from Cell Signaling Technology (Beverly, MA); Rabbit anti-PGP9.5 (NE1013) polyclonal antibody was obtained from Merck Millipore (Billerica, MA); Rabbit anti-IL-31Rα (ab113498) polyclonal antibody and rabbit anti-PAK-interacting exchange factor alpha (α-PIX) mAb (ab184569) were obtained from Abcam (Cambridge, MA); Recombinant mouse IL-31 (rIL-31) protein (#210–31) was obtained from PeproTech (Rocky Hill, NJ); FRAX597 was obtained from Cayman Chemical (Ann Arbor, MI). All other chemicals and reagents were of the highest grade commercially available.

## Animal study

All the experiments were conducted in accordance with the guidelines of the National Institutes of Health, as well as the Ministry of Education, Culture, Sports, Science and Technology of Japan, and were approved by the Fukushima Medical University Animal Studies Committee. All efforts were made to minimize animal suffering, to reduce the number of animals used, and to utilize alternatives to *in vivo* techniques. Male Nc/Nga mice (12 weeks old; Japan SLC, Shizuoka, Japan) with spontaneous dermatitis were used as previously described [13,14,21]. The mice were housed at 21˚C with a 12:12-h light/dark cycle with free access to water and a commercial diet. After preliminary breeding for one week, the mice were divided into four groups and received no ointment treatment or an application of 100 mg of one of the following vaseline ointments to the skin of the ear, face, neck, and rostral back three times a week: vehicle (0.1% DMSO), 0.05% PQA-18, or 0.1% FK506 [21]. After four weeks of the ointment treatment, the animals were euthanized by cervical dislocation to collect skin samples, followed by fixation with 10% formalin. To collect normal skin and ganglion samples, male C57BL/6J mice (12 weeks old; CLEA Japan, Tokyo, Japan) were used. To prepare sensory neuron cultures from dissociated DRG, C57Bl/6J mice were euthanized by cervical dislocation, and DRGs from the lumbar, thoracic, and cervical regions were removed [15]. DRGs were trimmed of connective tissue and nerve roots and then treated with 3 mg/ml collagenase for 2 h and then 0.25 mg/ml trypsin for 30 min. DRGs were triturated for dissociation to prepare a single-cell suspension. The DRG cells were plated onto culture dishes coated with poly-L-lysine (BD Biosciences) and laminin (BD Biosciences). The cells were cultivated in Neurobasal medium supplemented with 2% B27-supplement, 1% penicillin/streptomycin, 500 μM glutamine. DRG cells were stimulated with 100 ng/ml rIL-31 for 3 days in the absence or presence of PQA-18 at 100 nM.

## Immunohistochemistry

Skins were embedded in Tissue Tek (Sakura Finetek, Torrance, CA), frozen on dry ice, and stored at -80˚C. Cryostat sections (20 μm in thickness) were prepared, air-dried, and fixed in 10% neutral formaldehyde solution for 10 min [22]. Sections were blocked with 5% swine serum (Vector Laboratories, Burlingame, CA) and stained with rabbit anti-PGP9.5 antibody and rabbit anti-IL-31Rα antibody. Rabbit antibodies were detected with anti-rabbit IgG conjugated with Alexa Fluor 488 (Thermo Fisher Scientific). The sections were further mounted with VECTASHIELD (Vector Laboratories) and analyzed with an FV1000-D confocal microscope (Olympus, Tokyo, Japan). PGP9.5-positive or IL-31Rα-positive areas were measured in 10 different visual fields per section, which were randomly chosen in a blinded fashion. PGP9.5-positive or IL-31Rα-positive nerve fibers were quantified with a computer-assisted imaging program (ImageJ, 1.47V, US National Institutes of Health).

## Cell cultures

Mouse neuroblastoma Neuro2A cells (passage numbers 6–10, CCL-131: American Type Culture Collection, Manassas, VA) were cultivated in growth medium consisting in RPMI1640 supplemented with 10% (v/v) heat-inactivated fetal bovine serum (FBS, Sigma-Aldrich) in a humidified atmosphere of 5% $CO_2$ and 95% air at 37˚C. Human neuroblastoma SH-SY5Y cells (passage numbers 4–10, CRL-2266: American Type Culture Collection) were cultivated in growth medium consisting in a 1:1 mixture of MEM/Ham's F-12 supplemented with 10% (v/v) heat-inactivated FBS [22]. For IL-31 pathway analysis, Neuro2A cells were treated with PQA-18 and FRAX597 at different concentrations from 1 to 1000 nM for 30 min in the absence or presence of anti-IL-31Rα antibody at 50 ng/ml.

## Immunoblotting

Neuro2A cells and SH-SY5Y cells were solubilized in lysis buffer (PBS, pH 7.4, 1% *n*-dodecyl-β-D-maltoside [DDM], 1 mM Na$_3$VO$_4$) containing aprotinin (10 μg/ml), leupeptin (10 μg/ml), and phenylmethylsulfonyl fluoride (1 mM) [22]. After incubating on ice for 15 min, the lysates were clarified by centrifugation at 12,000 *g* for 15 min. After protein determination by a Bio-Rad protein assay reagent (Bio-Rad Laboratories, Hercules, CA), the supernatants (20 μg) were subjected to SDS-PAGE and the proteins were transferred to PVDF filter membranes (Millipore, Billerica, MA). The membranes were blocked with 5% non-fat dry milk in Tris-buffered saline containing 0.05% Tween 20 and incubated with primary antibodies. Blots were probed with goat anti-mouse IgG antibody or anti-rabbit IgG antibody coupled to HRP (Bio-Rad Laboratories), and the positive signals were visualized by ECL (PerkinElmer, Waltham, MA). Band intensities were quantified using Image J software.

## Immunocytochemistry

Cultured DRG cells and Neuro2A cells growing on glass coverslips were fixed with 10% neutral formaldehyde solution for 15 min at room temperature [22]. The cells were permeabilized with 0.1% Triton X-100 in PBS containing 5% swine serum for 1 h at room temperature and incubated with the primary antibody overnight at 4˚C. The cells were then reacted with anti-rabbit IgG antibody conjugated with Alexa Fluor 488 (Thermo Fisher Scientific) and anti-mouse IgG antibody conjugated with Alexa Fluor 546 (Thermo Fisher Scientific) for 1 h at room temperature, and observed under a confocal laser-scanning microscope system, FV-1000D (Olympus). Nerve fiber length of IL-31Rα-expressed DRG neurons was quantified with a computer-assisted imaging program (ImageJ) in 10 different visual fields per well, which were randomly chosen in a blinded fashion.

## RNA interference

The silencer select pre-designed short interfering RNA (siRNA) for mouse PAK2 (s104751) was obtained from Thermo Fisher Scientific. The scramble sequence for the control (5′−AGGUAGU GUAAUCGCCUUGdTdT−3′) was designed as previously described [21]. To achieve gene silencing, Neuro2A cells were transfected with the 80 nM siRNA for 24 h using the Neon Transfection System (Thermo Fisher Scientific) according to the manufacturer's recommended protocol. In our previous study, we confirmed over 90% transfection efficiency for siRNA [21].

## Neurite outgrowth assay

Neuro2A cells were plated at a density of 5 x 10$^4$ cells per well in a six-well plate and grown for 6 h in RPMI1640 with 10% FBS and then incubated with RPMI1640 with 0.1% FBS for 16 h prior to any treatment. The cells were treated with rIL-31 at 100 ng/ml, anti-IL-31Rα antibody at 50 ng/ml, and PQA-18 or FRAX597 at different concentrations from 1 to 1000 nM for 48 h. The cells were fixed with 10% formalin solution for 15 min and observed with a microscope. Pictures of 10 random fields/condition were taken per well. The cells with projections of a length at least two times greater than the cell diameter were scored as positive for neurite outgrowth [23]. Neurite outgrowth was quantified as the percentage of cells with neurites of total cells in the fields of corresponding condition.

## LC/MS/MS

The lysates of Neuro2A cells treated without or with PQA-18 at 100 nM for 30 min in the absence or presence of anti-IL-31Rα antibody at 50 ng/ml were clarified by centrifugation at

100,000 g for 15 min. To immunoprecipitate PAK2, the supernatants were incubated with PAK2 antibody-conjugated beads for 2 h, and washed with washing buffer (PBS, pH 7.4, 0.05% DDM) [24]. The precipitated proteins were reduced with dithiothreitol (8.3 mM) at 65˚C for 10 min, alkylated with iodoacetamide (14.5 mM) at room temperature for 30 min in the dark, and then digested with trypsin (1:40) at 30˚C for 12 h. The resulting peptides were purified on a C18 spin column (Pierce), and then dried almost completely in a vacuum centrifuge, and resuspended in 20 μL of 0.1% formic acid in water for LC/MS/MS. Liquid chromatography was performed on an Easy nanoLC II system (Thermo Fisher Scientific) coupled to an Orbitrap Elite mass spectrometer (Thermo Fishier Scientific). The Proteome Discoverer™ 1.4 software (Thermo Fishier Scientific) was used to generate the peak lists of all acquired MS/MS spectra, and these were then were automatically searched against the human SWISSPROT protein sequence database using the SEQUEST searching program (Thermo Fishier Scientific). To confirm the MS results, the precipitated proteins were also blotted with anti-α-PIX antibody and anti-PAK2 antibody.

## Data analysis

The statistical significance of differences was determined using the one-way analysis of variance with Turkey-Kramer post-hoc comparisons. Data are expressed as means and SD ([**], $p < 0.01$; [*], $p < 0.05$, as compared with control: [##], $p < 0.01$; [#], $p < 0.05$, [$$], $p < 0.01$, as compared with rIL-31 or anti-IL-31R antibody-treated group).

## Results

### Alleviation of excessive cutaneous nerve fiber density by PQA-18 ointment

To investigate the effect of PQA-18 on the pruritus in AD, we examined the sensory nerve fiber in lesional skin and treated skin from Nc/Nga mice. Nc/Nga mice (13 weeks of age) were divided into four groups (nine animals in each group) then treated with or without the ointment containing vehicle (DMSO), PQA-18, or FK506. FK506 was used as positive control for improvement of dermatitis [21]. Nerve fiber density was analyzed using immunohistochemistry to visualize protein gene product 9.5 (PGP9.5)-positive nerve fibers. PGP9.5 is known as a peripheral sensory nerve marker [15]. As shown in Fig 1A, treatment with PQA-18 ointment showed a significant decrease in the PGP9.5-positive nerve fiber density in treated skin as compared with that in lesional- or vehicle-treated skin in Nc/Nga mice. FK506 ointment did not significantly affect nerve fiber density. A recent study has demonstrated that the IL-31 pathway is associated with AD and develops a skin sensory neuron network to induce pruritus [15]. Thus, we further examined the effect of PQA-18 on IL-31Rα-positive nerve fibers in lesional skin and treated skin from Nc/Nga mice. Treatment with PQA-18 ointment also showed a significant decrease in the IL-31Rα-positive nerve fiber density in treated skin compared with that in lesional- or vehicle-treated skin in Nc/Nga mice (Fig 1B). FK506 ointment did not significantly affect IL-31Rα-positive fiber density. These results suggest that PQA-18 ointment, but not FK506 ointment, alleviates excessive cutaneous nerve fiber density in the skin of Nc/Nga mice.

### Inhibition of IL-31-induced sensory nerve development by PQA-18

To investigate the effect of PQA-18 on sensory nerve outgrowth, we examined the morphology of DRG neurons prepared from C57BL6J mice. Sensory nerve fiber was analyzed using immunocytochemistry to visualize βIII-tubulin- and IL-31Rα-positive nerve fibers. As shown in Fig 2, treatment with rIL-31 significantly enhanced βIII-tubulin-positive nerve fiber development

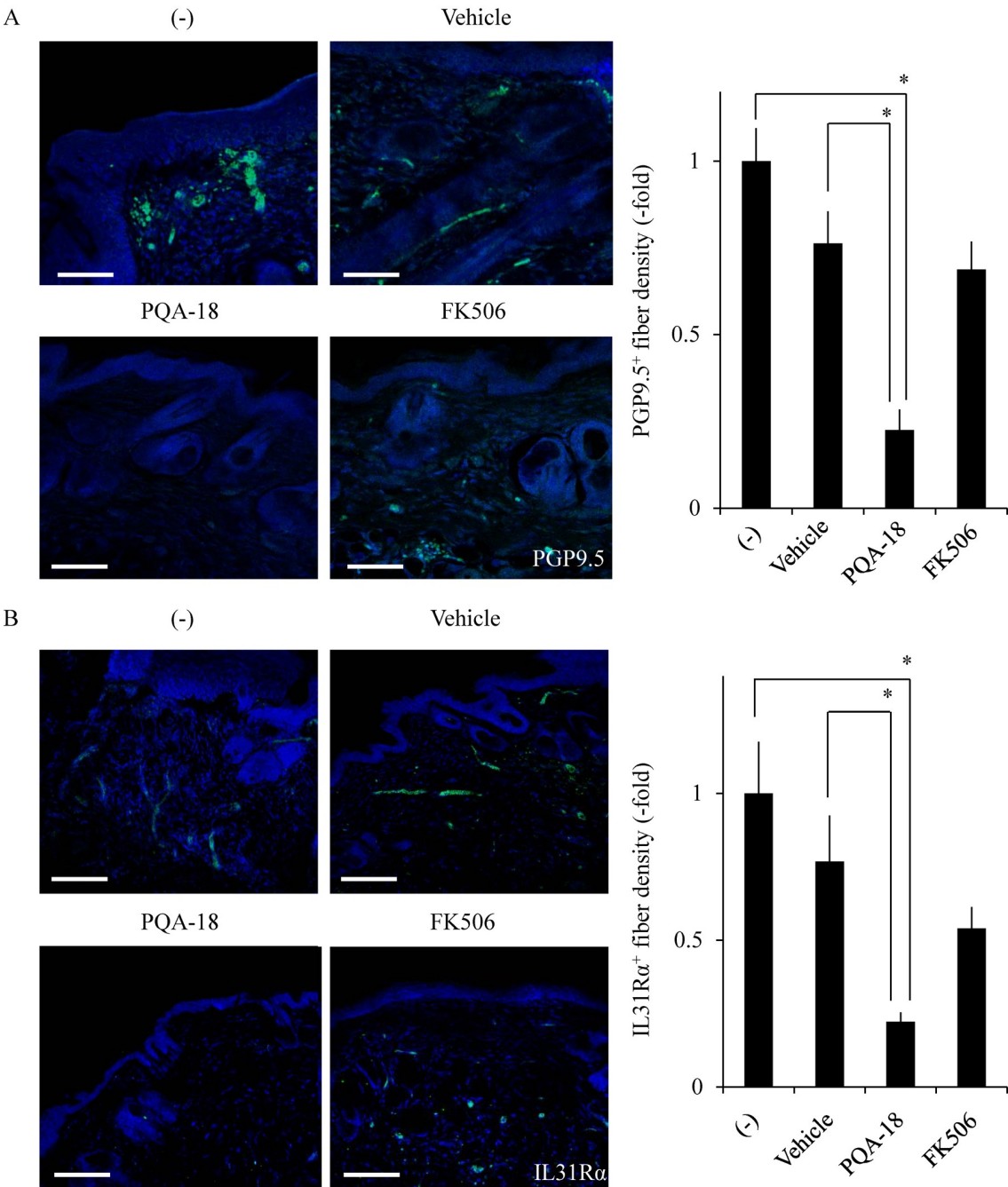

**Fig 1. Suppression of cutaneous nerve fiber density by PQA-18.** Sections were prepared from skin samples of Nc/Nga mice treated with or without vaseline ointment containing either vehicle, PQA-18 or FK506, and stained by anti-PGP9.5 antibody (green) (A) or the anti-IL-31Rα antibody (green) (B) and Hoechst33342 (blue; Nuclei). Scale bar: 100 μm. The representative images are shown on the left, and quantitative data of the number of nerve fiber are shown on the right. Data are pooled from three independent experiments with nine mice per group and shown as mean and SD. *$p < 0.05$, as compared with lesioned and vehicle (one-way ANOVA/Tukey-Kramer post-hoc comparisons).

as compared with vehicle treatment in IL-31Rα-expressed DRG neurons. PQA-18 significantly suppressed the IL-31-induced development, while PQA-18 alone did not affect normal sensory

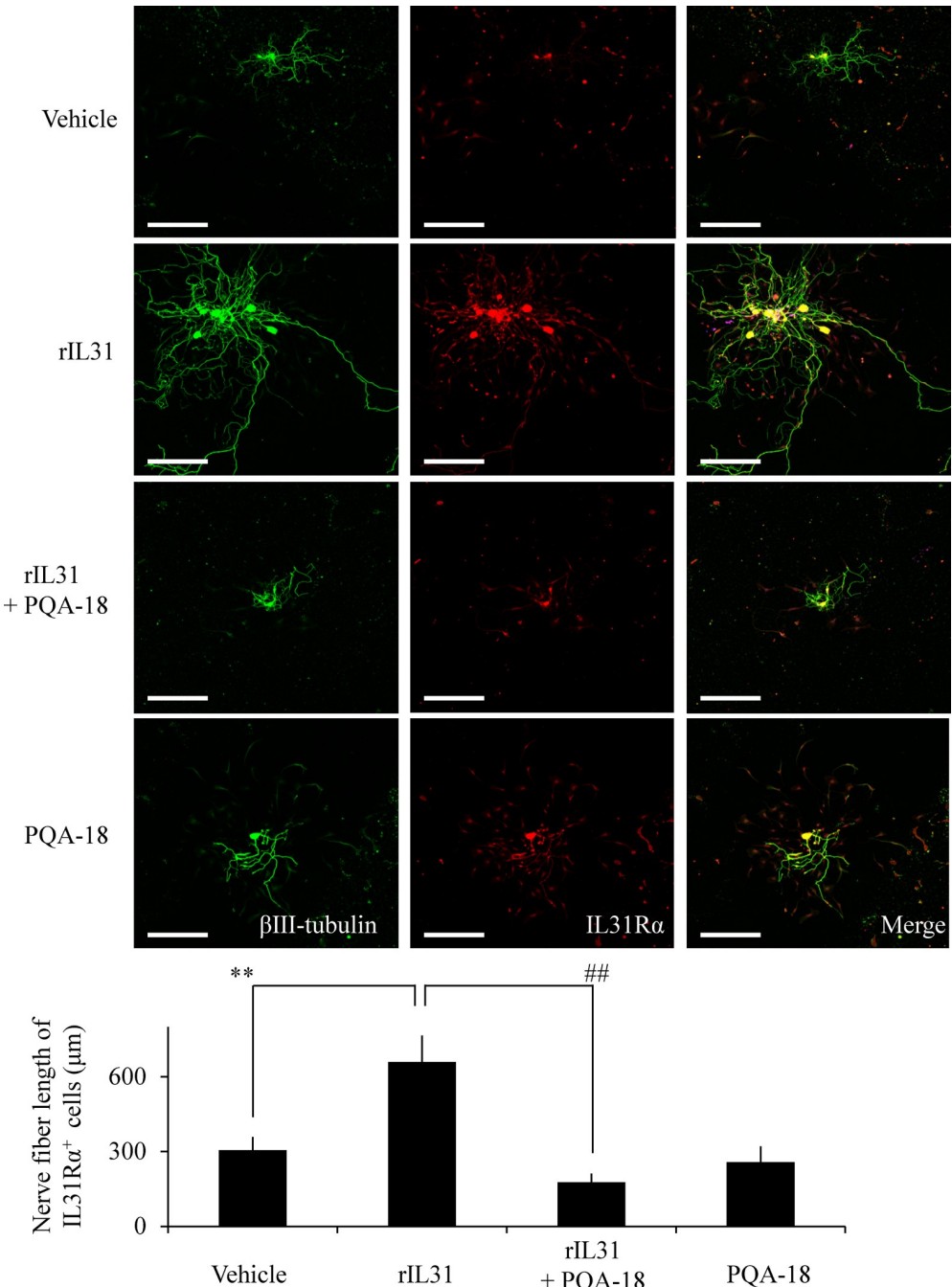

**Fig 2. Inhibition of IL-31-induced sensory nerve fiver outgrowth by PQA-18.** DRG cells were treated with or without rIL-31 in the absence or presence of PQA-18, and the cell morphology were examined by immunocytochemistry with anti-βIII-tubulin (green) and IL-31Rα (red) antibodies. Scale bar: 300 μm. The representative micrographs are shown (upper), and the quantitative data of the neurite outgrowth are shown (lower). Data are pooled from three independent experiments and shown as mean and SD. $^{**}p < 0.01$ as compared with control; $^{##}p < 0.01$ as compared with the rIL-31-treated group (one-way ANOVA/Tukey-Kramer post-hoc comparisons).

nerve development. These results suggest that PQA-18 inhibits IL-31-induced sensory nerve outgrowth.

## Inhibition of the IL-31 pathway by PQA-18

The above results led us to examine the effect of PQA-18 on the IL-31 pathway, using *in vitro* models of neurons; mouse Neuro2A and human SH-SY5Y neuroblastoma cells. We examined the expression of IL-31Rα in Neuro2A cells and SH-SY5Y cells by immunoblotting analysis with anti-IL-31Rα antibody. Mouse skin and ganglion samples were used as a positive control. We detected IL-31Rα in the lysate of Neuro2A cells, but not in SH-SY5Y cells (Fig 3A). Furthermore, immunocytochemistry analysis showed that Neuro2A cells express IL-31Rα and PGP9.5 (Fig 3B). In order to investigate its receptor function in Neuro2A cells, we examined the effect of rIL-31 and anti-IL-31Rα antibody on phosphorylation of STAT3. As shown in Fig 3C, the phosphorylation of STAT3 was enhanced by treatment with rIL-31 in a dose-dependent manner. Similar results were obtained using anti-IL-31Rα antibody (Fig 3D), indicating that the anti-IL-31Rα antibody also serves as a specific activator for IL-31R. These results suggest that Neuro2A cells functionally express IL-31R.

We examined the effect of PQA-18 on anti-IL-31Rα antibody-induced phosphorylation of STAT3 and upstream kinase JAK2 in Neuro2A cells. These phosphorylations were significantly enhanced by anti-IL-31Rα antibody. The enhancements significantly inhibited by treatment with PQA-18 in a dose-dependent manner (Fig 4), indicating that PQA-18 inhibits IL-31 pathway. Since PAKs is involved in cytokine signaling pathway [25] and PQA-18 is a PAK2 inhibitor [21], we further examined the effect of PQA-18 on activation of PAK2 in Neuro2A cells. PQA-18 also significantly inhibited anti-IL-31Rα antibody-induced phosphorylation of PAK2 (Fig 4). These results suggest that PQA-18 inhibits the IL-31 pathway through attenuation of PAK2 activation.

## Inhibition of IL-31-induced neurite outgrowth by PQA-18

We examined the effect of PQA-18 on IL-31-stimulating neurite outgrowth in Neuro2A cells. As shown in Fig 5, treatment with anti-IL-31Rα antibody significantly increased the number of neurite-positive cells in Neuro2A cells, while that increase was significantly inhibited by PQA-18 in a dose-dependent manner, suggesting that PQA-18 inhibits IL-31-induced neurite outgrowth in Neuro2A cells.

Based on the above results, selective group I PAK inhibitor FRAX597 was used to understand the role of PAK2 in neurite outgrowth [26,27]. Treatment with FRAX597 significantly inhibited anti-IL-31Rα antibody-induced phosphorylation of PAK2 (Fig 6A) and neurite outgrowth (Fig 6B and 6C) in a dose-dependent manner in Neuro2A cells, suggesting the involvement of PAK2 in IL-31-induced neurite outgrowth.

The effect of PAK2 siRNA on the neurite outgrowth was tested to confirm the PAK2 requirement for the IL-31-stimulating nerve fiber outgrowth. We confirmed that the PAK2 expression level is significantly suppressed by treatment with PAK2 siRNA in Neuro2A cells in a dose-dependent manner as compared with the control siRNA (Fig 7A). When the cells were introduced with PAK2 siRNA and then treated with either rIL-31 or anti-IL-31Rα antibody, the number of neurite-positive cells was significantly reduced as compared with control cells (Fig 7B). Furthermore, rIL-31- or anti-IL-31Rα antibody-induced phosphorylation of STAT3 was also significantly reduced by treatment with PAK2 siRNA (Fig 7C). These results suggest an indispensable role for PAK2 in IL-31-induced neurite outgrowth and STAT3 phosphorylation.

## Prevention of IL-31-induced formation of PAK2 activation complex by PQA-18

Several scaffold proteins including G protein-coupled receptor kinase interactor (GIT) and PIX are involved in PAK2 activation mechanism [28–30]. To understand the molecular

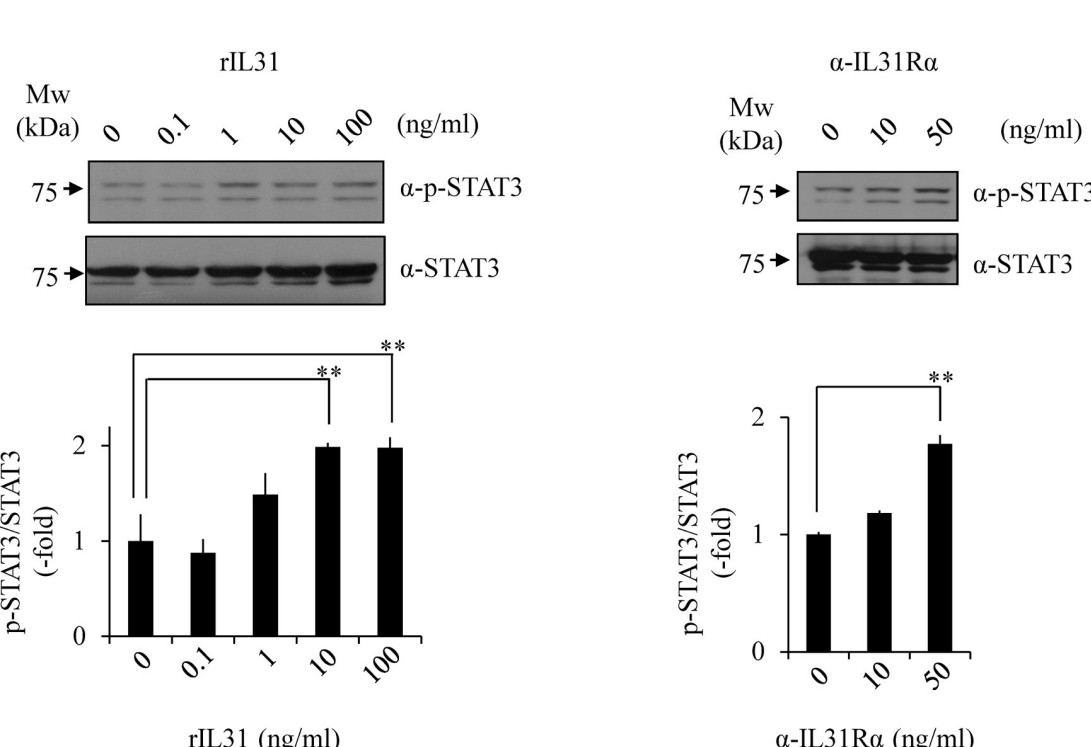

**Fig 3. Functional expression of IL-31R in Neuro2A cells.** The expression of IL-31Rα was determined by immunoblotting with anti-IL-31Rα antibody and anti-β-actin antibody (loading control) (A). The expression of IL-31Rα was analyzed by immunocytochemistry with anti-IL-31Rα antibody (green) or anti-PGP9.5 antibody (green) and Hoechst33342 (blue; Nuclei) (B). The representative micrographs are shown. Scale bar, 100 μm. Phosphorylated STAT3 at Tyr705 was analyzed by immunoblotting. The cells were treated with rIL-31 (C) or anti-IL-31Rα antibody (D) at indicated concentrations, and the cell lysates were examined by immunoblotting with indicated antibodies. The representative images are shown (upper) and the quantitative data of the ratios of phosphorylated STAT3 versus STAT3 are shown (lower). Data are pooled from three independent experiments and shown as mean and SD. $^{**}p < 0.01$, as compared with control (one-way ANOVA/Tukey-Kramer post-hoc comparisons).

mechanisms underlying inhibition of PAK2 by PQA-18, we examined PAK2 complex components by LC/MS/MS. PAK2 complexes were purified from cell lysates prepared from Neuro2A

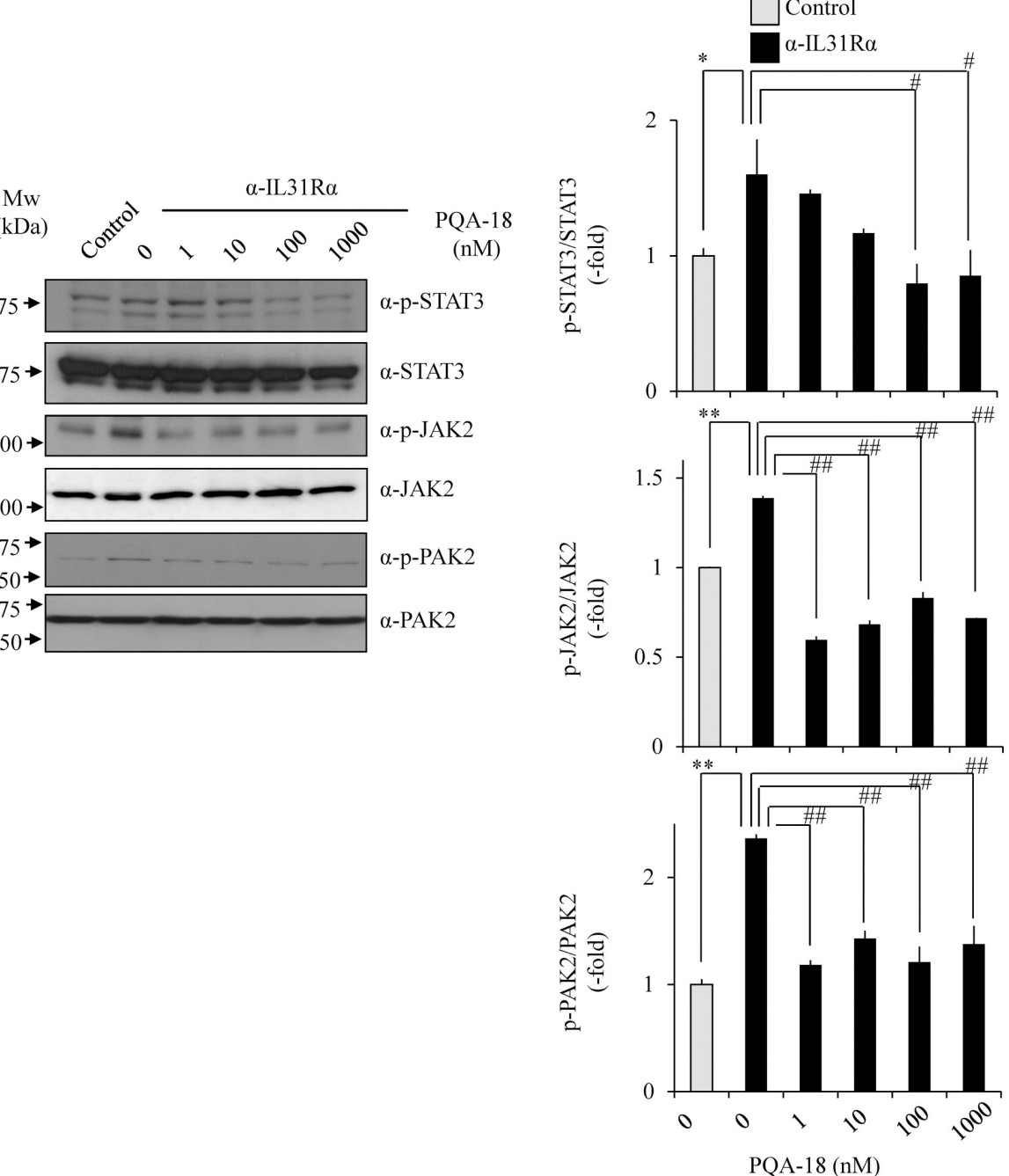

**Fig 4. Inhibition of IL-31 pathway by PQA-18.** Phosphorylated STAT3 at Tyr705, JAK2 at Tyr1008, and PAK2 at Ser141 were analyzed by immunoblotting. Neuro2A cells were treated with anti-IL-31Rα antibody in the absence or presence of PQA-18 at indicated concentrations, and the cell lysates were examined by immunoblotting with indicated antibodies. The representative images are shown on the left, and the quantitative data of the ratios of phosphorylated STAT3 versus STAT3, phosphorylated JAK2 versus JAK2, and phosphorylated PAK2 versus PAK2 are shown on the right. Data are pooled from three independent experiments and shown as mean and SD. $^{**}p < 0.01$, $^{*}p < 0.05$ as compared with control; $^{\#\#}p < 0.01$, $^{\#}p < 0.05$ as compared with anti-IL-31Rα antibody-treated group (one-way ANOVA/Tukey-Kramer post-hoc comparisons).

cells treated with PQA-18 in the absence or presence of anti-IL-31Rα antibody using anti-PAK2 antibody conjugated beads. When the purified PAK2 complex was digested with trypsin

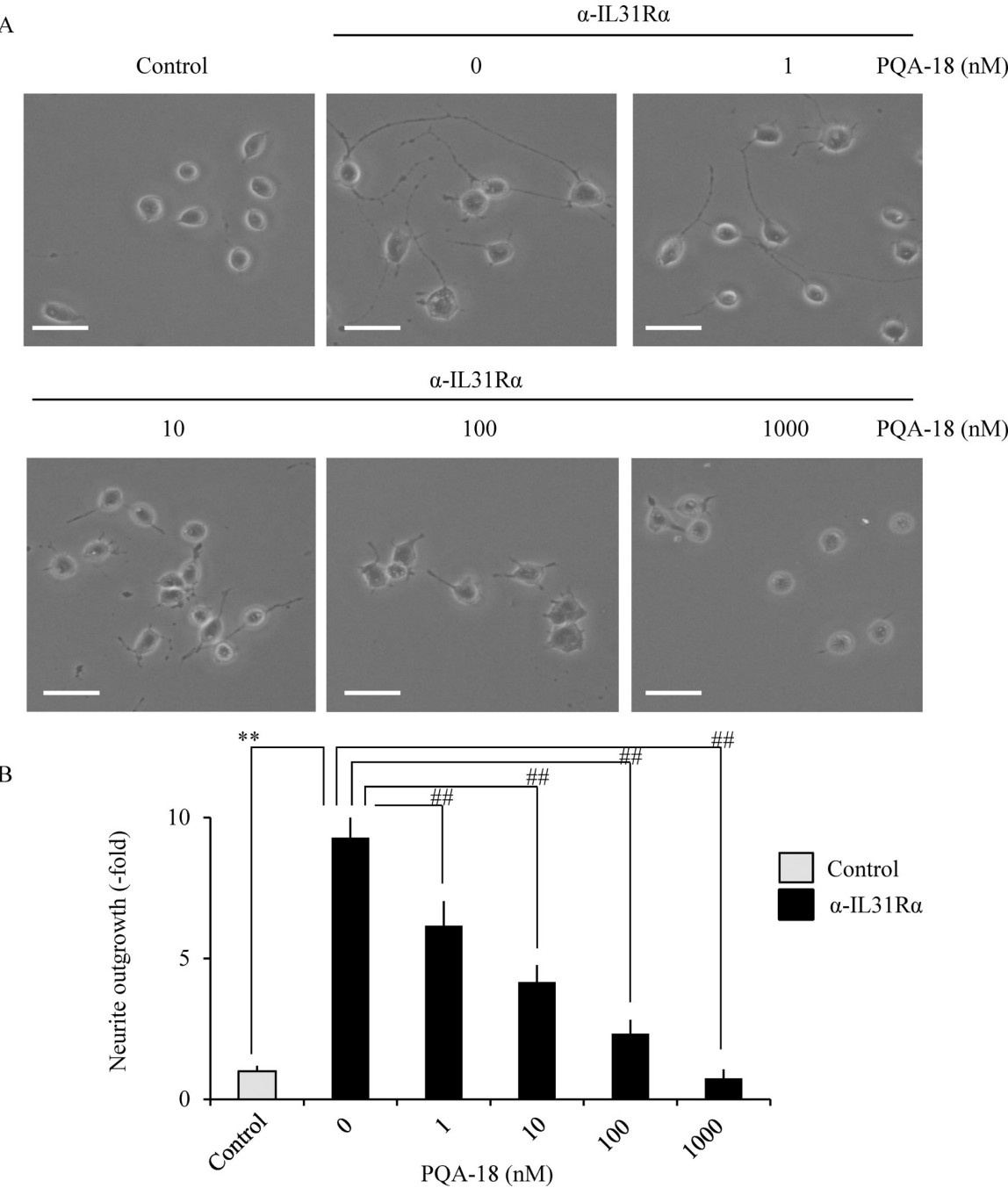

**Fig 5. Inhibition of IL-31-induced neurite outgrowth by PQA-18.** Neuro2A cells were treated with anti-IL-31Rα antibody in the absence or presence of PQA-18 at indicated concentrations, and the cell morphology was examined by microscopy. Neurites were defined as a process with lengths equivalent to one diameter of a cell body. The percentage of neurite-bearing cells was calculated from the total number of counted cells. The representative micrographs are shown (A), and the quantitative data of the neurite outgrowth are shown (B). Scale bar: 20 μm. Data are pooled from three independent experiments and shown as mean and SD. **$p < 0.01$ as compared with control; ##$p < 0.01$ as compared with anti-IL-31Rα antibody-treated group (one-way ANOVA/Tukey-Kramer post-hoc comparisons).

and the resulting peptides were analyzed by LC/MS/MS, 12 proteins could be identified in each sample (Table 1). Unique peptides for GIT1, GIT2, α-PIX and β-PIX were identified as

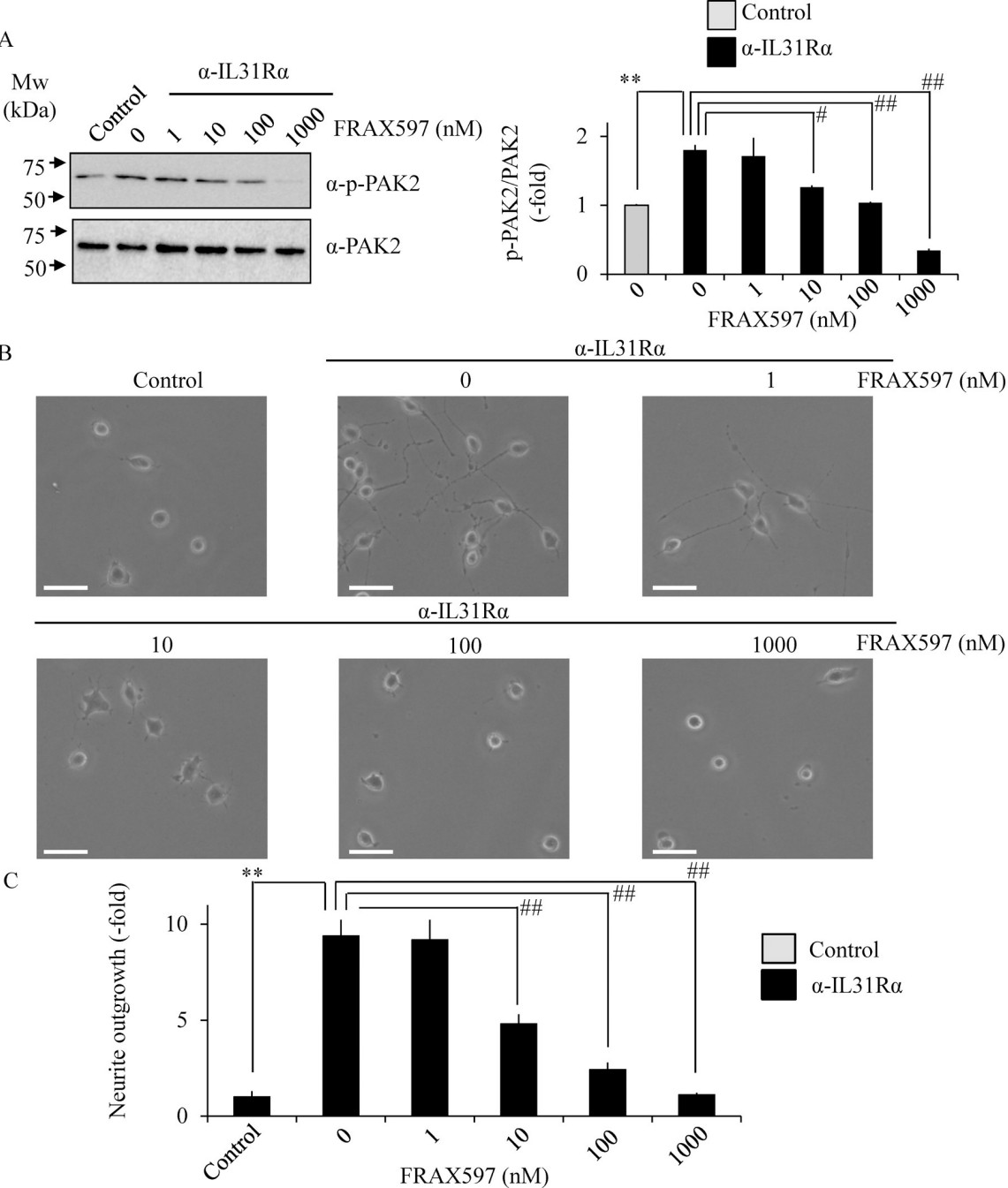

**Fig 6. Inhibition of IL-31-induced neurite outgrowth by FRAX597.** Neuro2A cells were treated with anti-IL-31Rα antibody in the absence or presence of FRAX597 at indicated concentrations. Phosphorylated PAK2 at Ser141 were analyzed by immunoblotting (A). The representative images are shown on the left, and the quantitative data of the ratios of phosphorylated PAK2 versus PAK2 are shown on the right. Neurites were defined as a process with lengths equivalent to one diameter of a cell body. The percentage of neurite-bearing cells was calculated from the total number of counted cells. The representative micrographs are shown (B), and the quantitative data of the neurite outgrowth are shown (C).Scale bar: 20 μm. Data are pooled from three independent experiments and shown as mean and SD. $^{**}p < 0.01$ as compared with control; $^{##}p < 0.01$, $^{#}p < 0.05$ as compared with anti-IL-31Rα antibody-treated group (one-way ANOVA/Tukey-Kramer post-hoc comparisons).

components of the PAK2 activation complex. On the other hand, these peptides were not detected in normal rabbit IgG-beads as a negative control. As shown in Fig 8A, the peak areas

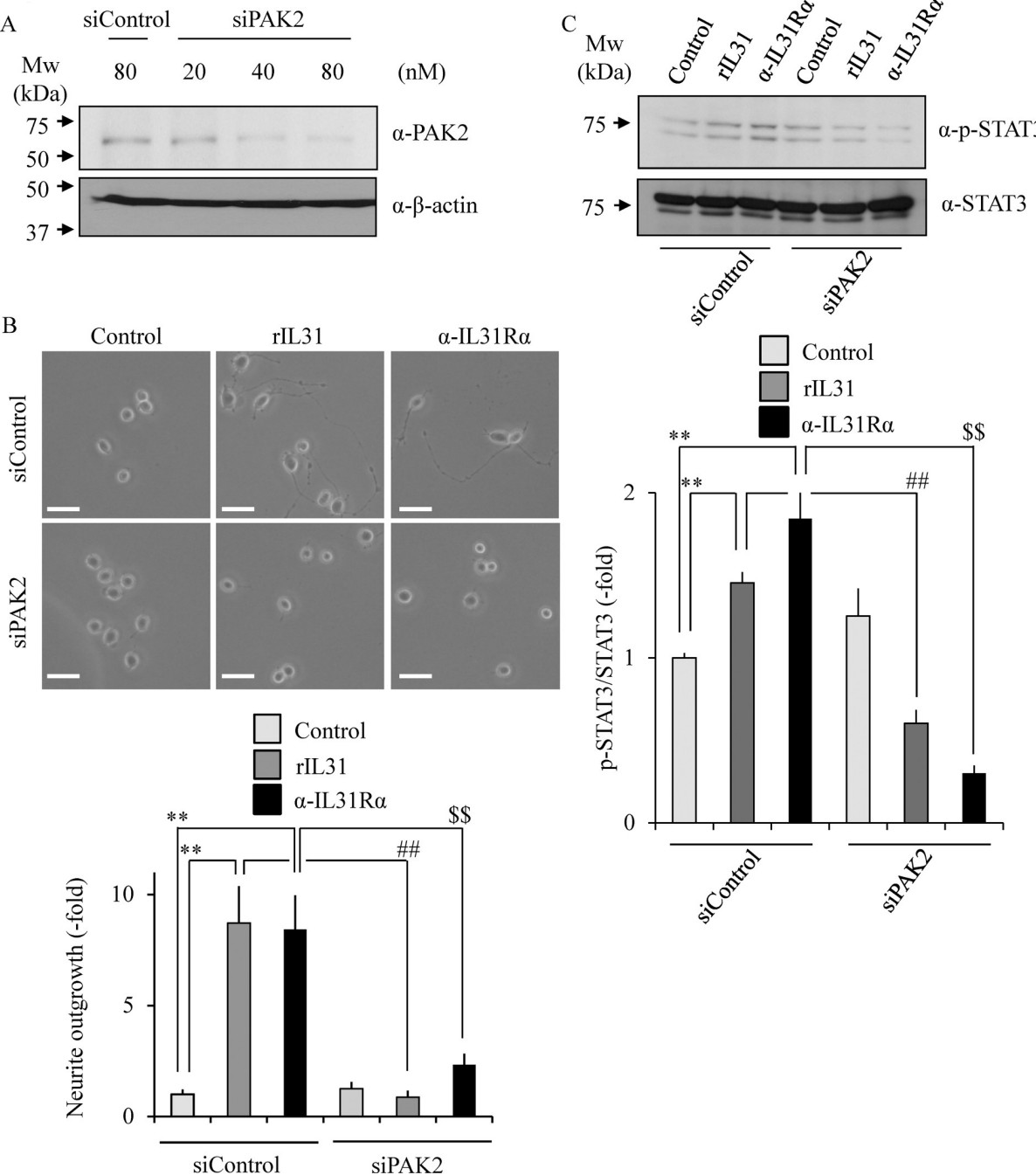

**Fig 7. Requirement of PAK2 for IL-31-induced neurite outgrowth and STAT3 phosphorylation.** Neuro2A cells were transfected with siRNA for PAK2 or control siRNA. PAK2 expression was determined by immunoblotting (A). Representative images of Neuro2A cells after transfection with siRNA for PAK2 or control siRNA. Neurites were defined as a process with lengths equivalent to one diameter of a cell body (B). The percentage of neurite-bearing cells was calculated from the total number of counted cells. Scale bar: 20 μm. Phosphorylated STAT3 at Tyr705 was analyzed by immunoblotting (C). Neuro2A cells were treated with rIL-31 or anti-IL-31Rα antibody in the absence or presence of siRNA for PAK2, and the cell lysates were examined by immunoblotting with indicated antibodies. The representative images are shown (upper), and the quantitative data of the ratios of phosphorylated STAT3 versus STAT3 are shown (lower). Data are pooled from three independent experiments and shown as mean and SD. $**p < 0.01$ as compared with control; $^{\#\#}p < 0.01$ as compared with rIL-31-treated group; $^{\$\$}p < 0.01$ as compared with anti-IL-31Rα antibody-treated group (one-way ANOVA/Tukey-Kramer post-hoc comparisons).

**Table 1. PAK2 interacting proteins as identified by LC/MS/MS.**

| Description | ΣCoverage | Number of Unique Peptides | | | |
|---|---|---|---|---|---|
| | | Control | PQA-18 | α-IL-31R | α-IL-31R+PQA-18 |
| Serine/threonine-protein kinase PAK2 | 55.15 | 93 | 80 | 103 | 86 |
| Rho guanine nucleotide exchange factor 7 β-PIX | 18.68 | 13 | 16 | 50 | 7 |
| Peripherin | 18.11 | 0 | 0 | 31 | 8 |
| H1 histone family, member X | 17.02 | 0 | 5 | 9 | 8 |
| Tubulin beta-5 chain | 15.09 | 0 | 3 | 0 | 9 |
| ARF GTPase-activating protein GIT1 | 14.16 | 12 | 10 | 19 | 15 |
| Rho guanine nucleotide exchange factor 6 α-PIX | 9.73 | 0 | 1 | 5 | 1 |
| ARF GTPase-activating protein GIT2 | 4.38 | 0 | 0 | 10 | 0 |
| Casein kinase II subunit alpha-interacting protein | 3.61 | 0 | 0 | 1 | 0 |
| Protein transport protein sec16 | 3.35 | 0 | 2 | 0 | 3 |
| Alpha-1,4-N-acetylglucosaminyltransferase | 2.93 | 0 | 0 | 1 | 0 |
| Alpha-actinin-1 | 2.47 | 0 | 0 | 6 | 6 |

for GIT2, α-PIX, and β-PIX were significantly increased when Neuro2A cells were treated with anti-IL-31Rα antibody. The increase of the peak areas for GIT2 and α-PIX was significantly inhibited by treatment with PQA-18. To confirm the above results, we further examined PAK2 interaction with α-PIX by immunoprecipitation assay. Treatment with anti-IL-31Rα antibody significantly increased PAK2 interaction with α-PIX, while that increase was significantly inhibited by treatment with PQA-18 (Fig 8B). These results suggest that PQA-18 inhibits IL-31-induced PAK2 activation through preventing formation of the PAK2 activation complex with GIT2 and α-PIX.

## Discussion

In this study, we demonstrated that PQA-18 suppresses sensory nerve outgrowth through inhibition of the IL-31 pathway. IL-31 promotes the development of sensory nerve fibers by activating IL-31R [15]. We observed that neurite outgrowth is significantly promoted by treatment with rIL-31 and anti-IL-31Rα antibody in vitro experiments using DRG neurons (Fig 2) and neuronal model Neuro2A cells (Figs 3 and 5), which express IL-31R. Moreover, PQA-18 treatment suppressed the neurite outgrowth in the two independent cells (Figs 2 and 5). The IL-31 pathway is known to activate the intracellular JAK/STAT pathway [9,15]. JAK has four known molecules, JAK1, JAK2, JAK3, and tyrosine kinase 2, and interacts with various cytokine receptors [31]. After autophosphorylation by cytokine receptor activation, JAKs phosphorylate STATs, which promote dimer formation and nuclear import, and regulate downstream target gene transcription [32]. IL-31 is thought to mainly promote phosphorylation of STAT3 though JAK1 or JAK2 activation [5]. A recent study revealed that phosphorylation of STAT3 promotes the expression of genes involved in neurite outgrowth in the sensory nervous system [15]. It has also been reported that neurite outgrowth is suppressed by pharmacological inhibition of STAT3 and a dominant negative form of STAT3 [15,33]. We observed that stimulation of IL-31R induces phosphorylation of JAK2 and STAT3 in Neuro2A cells (Fig 4). PQA-18 suppressed the IL-31R-induced phosphorylation of JAK2 and STAT3 (Fig 4). These results suggest that PQA-18 inhibits IL-31/JAK2/STAT3 pathway. Increased IL-31 levels and excessive cutaneous sensory innervation are observed in AD patients and AD models including Nc/Nga mice, leading to lowered the pruritus threshold and causing severe pruritus and dermatitis [12–14]. These symptoms are improved by application of anti-IL-31 antibody and anti-IL-31R-neutralizing antibody [5]. Hence, IL-31 pathway plays a crucial role

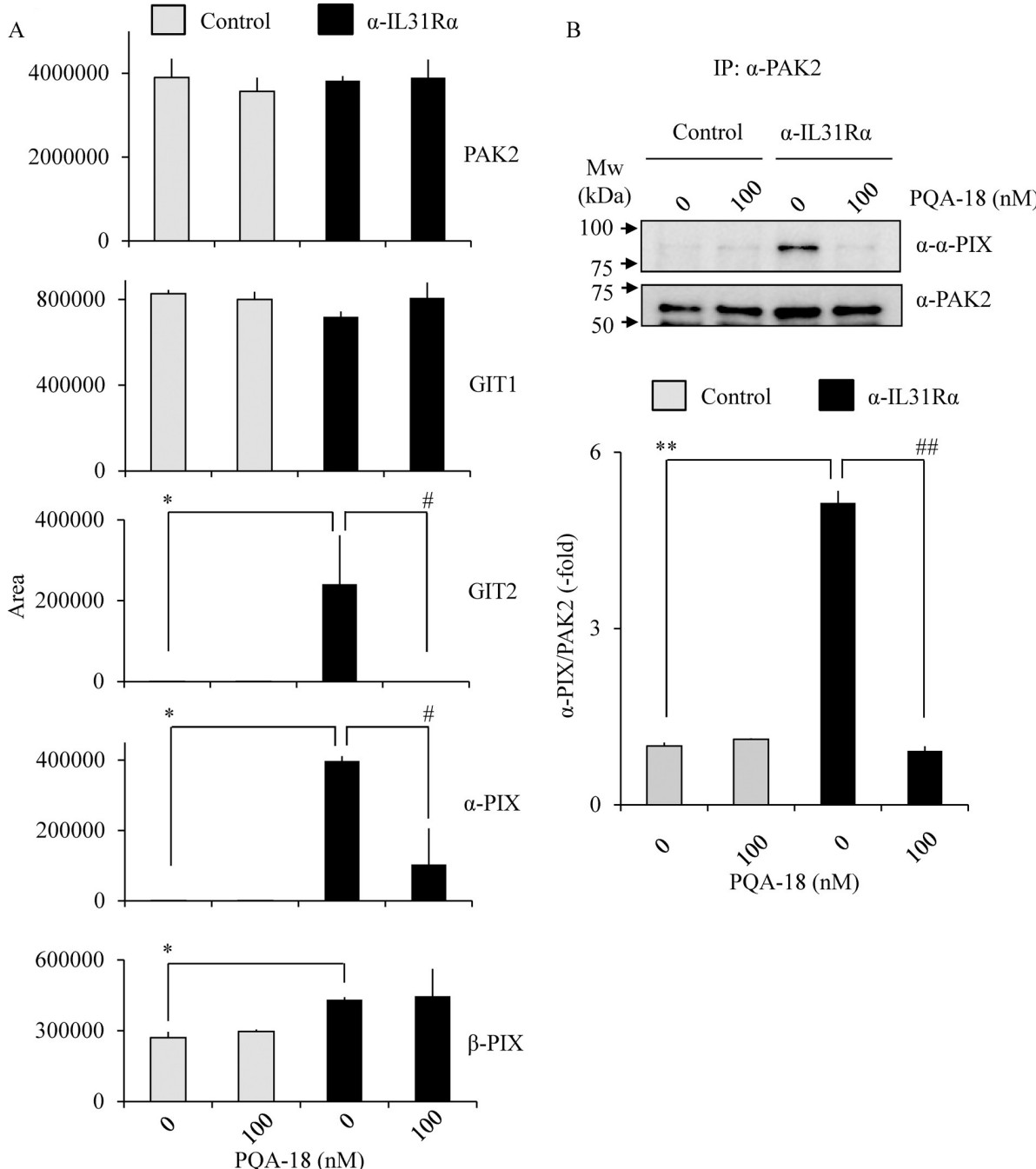

**Fig 8. Prevention of formation of PAK2 activation complex by PQA-18.** PAK2 interacting proteins as identified by LC/MS/MS (A). LC/MS/MS analysis was performed using tryptic peptides of purified PAK2 prepared from Neuro2A cells treated without or with PQA-18 in the absence or presence of anti- IL-31Rα antibody using anti-PAK2 antibody conjugated beads. The quantitative data of the peak area of the indicated proteins are shown. Immunoprecipitates were obtained from Neuro2A cells treated without or with PQA-18 in the absence or presence of anti- IL-31Rα antibody using anti-PAK2 antibody conjugated beads, and immunoblotted with anti-α-PIX antibody or anti-PAK2 antibody (B). The representative images are shown (upper), and the quantitative data of the ratios of α-PIX versus PAK2 are shown (lower). Data are pooled from three independent experiments and shown as mean and SD. $^{**}p < 0.01$, $^{*}p < 0.05$ as compared with control; $^{##}p < 0.01$, $^{#}p < 0.05$ as compared with anti-IL-31Rα antibody-treated group (one-way ANOVA/Tukey-Kramer post-hoc comparisons).

in development of cutaneous sensory innervation *in vivo*. Indeed, we found that administration of the PQA-18 ointment reduces cutaneous sensory nerve density in Nc/Nga mice (Fig 1).

Taken together, these results suggest that PQA-18 suppresses sensory nerve outgrowth through inhibition of the IL-31 pathway both *in vitro* and *in vivo*. We speculate that PQA-18 could improve the pruritus of AD by suppression of the excessive sensory innervation in dermatitis.

Given that our previous finding of PQA-18 being a PAK2 inhibitor [21], it is conceivable that PAK2 also plays an important role in neurite outgrowth. PAKs, effector molecules of Rac and Cdc42, are serine/threonine kinases that phosphorylate multiple substrates, including those that are involved in cytoskeletal reorganization, cell proliferation, and survival [34]. The PAK family is divided into two groups (group I and group II) based on sequence and structural homology. PAK2, group I PAK, is expressed in neuronal cells and has been reported to be implicated in crucial neuronal functions including synapse formation [35]. We observed that IL-31R stimulation enhances phosphorylation of PAK2 and STAT3, and neurite outgrowth in Neuro2A cells. These enhancements were significantly inhibited by PQA-18 treatment (Figs 4 and 5). In addition, FRAX597 treatment also significantly inhibited IL-31R-induced phosphorylation of PAK2 and neurite outgrowth (Fig 6). These results suggest the involvement of PAK2 in neurite outgrowth system. Although the role of PAK2 in neurite outgrowth is not well understood, the group I PAK inhibitor IPA3 has been reported to inhibit STAT3 phosphorylation and neurite outgrowth [25,36]. We observed inhibition of IL-31-induced STAT3 phosphorylation and neurite outgrowth from analysis using Neuro2A cells that suppress PAK2 expression (Fig 7). These results indicate that PAK2 is required for IL-31-induced activation of STAT3 and neurite outgrowth. Thus, PAK2 may play a key role in regulation of sensory nerve outgrowth. Further analysis of the role of PAK2 in the development of cutaneous sensory nerves is needed to clarify the mechanism of pruritus development in AD.

Although the mechanism of PAK2 activation is not yet fully understood, understanding of the mechanism by which IL-31 activates PAK2 is important in understanding the IL-31 pathway. Group I PAKs exists in an inactive form by dimerization, but dissociates due to the interaction of activated Rac and cdc42, interacts with the PIX/GIT complex known as a scaffold protein, which promotes autophosphorylation due to high local concentrations of PAKs [28]. We have previously reported that autophosphorylation of PAK2 at Ser141 plays an important role in kinase activity [21,34]. From the analysis using the anti-phospho-Ser141 PAK2 antibody, we observed that the phosphorylated PAK2 at Ser141 is increased by the IL-31R activation (Fig 4). PQA-18 inhibited the phosphorylation of PAK2 at Ser141 (Fig 4), indicating that PQA-18 suppresses autophosphorylation of PAK2 induced by IL-31R. Furthermore, when the activated PAK2 complex was analyzed using LC/MS/MS, 12 types of unique peptides including α-PIX, β-PIX, GIT1 and GIT2 were detected (Table 1). Quantitative analysis of these peptides revealed the significant increased the interaction of GIT2, α-PIX and β-PIX with PAK2 by IL-31R stimulation (Fig 8). Among them, the amount of interaction of GIT2 and α-PIX with PAK2 was significantly decreased by PQA-18 treatment (Fig 8). We further confirmed that PQA-18 treatment significantly inhibits IL-31R-induced the interaction of PAK2 with α-PIX by immunoprecipitation assay (Fig 8). These results indicate that the interaction of PAK2 with the PIX/GIT complex plays an important role in the mechanism of PAK2 activation by the IL-31 pathway. PQA-18 might prevent the formation of PAK2 activation complexes by inhibiting GIT2 and α-PIX interactions, supporting previous our observation that PQA-18 inhibits PAK2 activity in non-competitive manner [21]. In future, functional analysis of PAK2 interacting components and the structural analysis of PAK2 activation complexes are necessary to clarify the IL-31 pathway.

Our observations demonstrated that PQA-18 at low concentrations (1–10 nM) strongly inhibits IL-31R-induced phosphorylation of PAK2 (Fig 4) but has a limited effect on neurite outgrowth (Fig 5). On the other hands, FRAX597 had a similar inhibitory dose response

between PAK2 phosphorylation and neurite outgrowth (Fig 6). The difference between the two compounds is that PQA-18 inhibits the PAK2 activation complex as described above (Fig 8), whereas FRAX597 inhibits PAK2 activity by the competitive inhibition of ATP [26,27]. Recent study has revealed that PAK-PIX interactions as well as STAT3 activation regulate neurite growth [37]. Interestingly, complete inhibition of PAK-PIX interaction suppressed neurite outgrowth, whereas partial inhibition promoted neurite outgrowth [37]. Therefore, it is possible that partial inhibition of PAK-PIX interaction may be involved in the attenuation of the inhibitory effect of neurite outgrowth by PQA-18 at low concentration. Further analysis of the physiological role of PAK-PIX interaction is required for understanding the mechanism underlying the regulation of neurite outgrowth by PQA-18.

The scratching behavior mechanically damages the skin, lowers the barrier function, enhances the inflammatory reaction by foreign antigens that have penetrated through the epidermis, and aggravates dermatitis and further enhances pruritus. Such vicious cycle of scratching, exacerbation of inflammation, and enhancement of the itch is called the itch-scratch-cycle, and is known to contribute to chronic AD [5,38]. Therefore, the development of an AD therapeutic drug that suppresses not only inflammation but also pruritus is highly desired. Currently, glucorticoids and calcineurin inhibitors are used as therapeutic agents for AD, and mainly target skin inflammation [39]. However, these drugs have serious side effects including adrenal failure, skin atrophy, neurotoxicity, nephrotoxicity, skin cancer, tumor growth, due to long-term administration [39]. Although immunosuppressant FK506 is also used to treat AD, it has been reported to cause not only nephrotoxicity and hepatotoxicity as adverse effects but also itching sensation [40,41]. We observed that application of PQA-18 ointment to Nc/Nga mice significantly improves excessive sensory nerve density and IL-31R expression, but FK506 ointment fail to improve it (Fig 1). Thus, it is conceivable that FK506 did not target the IL-31 pathway, resulting in weak therapeutic effects on pruritus, which is in line with previous study which showed that IL-31-induced scratching behavior is not inhibited by treatment with FK506 [42]. Recently, the development of a novel AD therapeutic drug targeting the IL-31 pathway has advanced. Nemolizumab, an IL-31R-neutralizing humanized antibody, is effective in improving pruritus in AD patients, but clinical studies have shown that exacerbation of dermatitis and peripheral edema are observed in some patients [43,44]. Given our previous findings that PQA-18 suppresses inflammatory cytokine production through inhibition of PAK2 in T cells and does not cause side effects such as renal damage, liver damage or tumor formation even with long-term systemic administration in mice [21], PQA-18 is considered to be a lead compound for the development of a useful therapeutic drug targeting both inflammation and pruritus for AD.

In conclusion, we have demonstrated that PQA-18 is an inhibitor of sensory nerve outgrowth. PQA-18 inhibits the IL-31 pathway by suppressing PAK2 activity, which in turn suppresses STAT3 activation. We have further demonstrated that PQA-18 exhibits an inhibitory effect on excessive cutaneous sensory nerve development in an AD model. These results suggest that PQA-18 may be a feasible lead compound for treatment of pruritus in AD.

## Supporting information

**S1 Fig. Full blot data of Fig 3A.**
(PDF)

**S2 Fig. Full blot data of Fig 3C.**
(PDF)

**S3 Fig. Full blot data of Fig 3D.**
(PDF)

**S4 Fig. Full blot data of Fig 4A.**
(PDF)

**S5 Fig. Full blot data of Fig 4B.**
(PDF)

**S6 Fig. Full blot data of Fig 4C.**
(PDF)

**S7 Fig. Full blot data of Fig 6A.**
(PDF)

**S8 Fig. Full blot data of Fig 7A.**
(PDF)

**S9 Fig. Full blot data of Fig 7C.**
(PDF)

**S10 Fig. Full blot data of Fig 8.**
(PDF)

**S1 Table. Minimal underlying data.**
(PDF)

## Author Contributions

**Conceptualization:** Masato Ogura, Yoshimi Homma.

**Data curation:** Masato Ogura, Toshiyuki Suzuki.

**Formal analysis:** Masato Ogura.

**Funding acquisition:** Masato Ogura, Yoshimi Homma.

**Investigation:** Masato Ogura, Kumiko Endo, Toshiyuki Suzuki.

**Methodology:** Masato Ogura.

**Project administration:** Masato Ogura.

**Resources:** Masato Ogura.

**Software:** Masato Ogura.

**Supervision:** Masato Ogura.

**Validation:** Masato Ogura.

**Visualization:** Masato Ogura.

**Writing – original draft:** Masato Ogura.

**Writing – review & editing:** Masato Ogura, Yoshimi Homma.

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
