## [Decision Letter · Decision Letter 0]

2 Oct 2020

PONE-D-20-23391

Prenylated quinolinecarboxylic acid compound-18 prevents sensory nerve fiber outgrowth through inhibition of the interleukin-31 pathway

PLOS ONE

Dear Dr. Ogura,

Thank you for submitting your manuscript to PLOS ONE. After careful consideration, we feel that it has merit but does not fully meet PLOS ONE’s publication criteria as it currently stands. Therefore, we invite you to submit a revised version of the manuscript that addresses the points raised during the review process.

Please find enclosed the expert reviews on your manuscript. Although the results presented are considered important in the field, better validation of the inhibitory effect of PQA-18 on PAK2 is required. In particular some key results should be compared with other PAK-specific inhibitors, which are available. There are issues with regard to the presentation of results that need to be addressed. I hope you will be able to carry out the additional experiments in the time requested.

We look forward to receiving your revised manuscript.

Kind regards,

Ed Manser, PhD

Academic Editor

PLOS ONE

Journal Requirements:

2. In your Methods section, please provide additional information on the animal research and ensure you have included details on : (1) methods of sacrifice (2) methods of anesthesia and/or analgesia, and (2) efforts to alleviate suffering.

Reviewers' comments:

Reviewer's Responses to Questions

**Comments to the Author**

1. Is the manuscript technically sound, and do the data support the conclusions?

Reviewer #1: Partly

Reviewer #2: Partly

2. Has the statistical analysis been performed appropriately and rigorously? 

Reviewer #1: Yes

Reviewer #2: Yes

3. Have the authors made all data underlying the findings in their manuscript fully available?

Reviewer #1: Yes

Reviewer #2: Yes

4. Is the manuscript presented in an intelligible fashion and written in standard English?

Reviewer #1: Yes

Reviewer #2: Yes

5. Review Comments to the Author

Reviewer #1: Yoshimi Homma’s lab had previously reported that PQA-18, a derivative of Polysphondylium pseudo-candidum isolated small molecule Ppc-1, acts as an immunosuppressant and a PAK2 kinase inhibitor, could alleviate atopic dermatitis in mouse model (ref. 21). In the current study, the authors are trying to understand the mechanism of action of PQA-18 in relieving the skin defects in atopic dermatitis.

The authors chose to examine the IL-31/IL-31R� pathway and provide data to show that PQA-18 decreased the cutaneous sensory nerve fibre density by inhibiting the expression of IL-31 receptor alpha in mouse skin. In cultured neuronal cells, PQA-18 inhibits IL-31 induced sensory nerve fibre outgrowth and neurite outgrowth by blocking IL-31R� activation of the PAK2/JAK2/STAT3 pathway. Lastly, PQA-18 selectively blocks PAK2 interaction with GIT2 and �-PIX, which are thought to activate PAK2.

That PQA-18 acts as a potent immunosuppressant and effectively improves the skin pruritus condition in mouse AD model are potentially important findings. Experimental data for that PQA-18 inhibits IL-31/IL-31R� pathway and prevents cutaneous nerve fibre growing is convincing. But the molecular mechanism described for PQA-18 inhibiting PAK2 kinase is less convincing. There are a several issues which the authors need to address.

Major points.

1. The authors believed that PAK2 is an important target of PQA-18 and based on MS data in Fig.7 they suggest that PQA-18 inhibited PAK2 activation by preventing its interaction with GIT2 / �-PIX. However more direct evidence is required to demonstrate PQA-18 blocks PAK2 interaction with �-PIX (GIT2 is likely indirect) using a direct binding assay.

2. Since both PAK1 and PAK2 are expressed in neuronal cells (PAK2 being ubiquitous) the authors should clarify if PQA-18 has similar effects on PAK1 kinase.

3. In Fig.2 (compare IL-31R� images of vehicle vs PQA-18) the data does not support the claim that PQA-18 inhibits IL-31Ra expression in cutaneous nerve fibres. Cf. as stated on page 2 “PQA-18 alleviates cutaneous nerve fibre density and the expression of IL-31 receptor α (IL-31Rα) in the skin of Nc/Nga mice.” and page 11 “These results suggest that PQA-18 ointment, but not FK506 ointment, alleviates excessive cutaneous nerve fibre density and expression of IL-31Rα in the skin of Nc/Nga mice”.

4. Do the authors think that PAK2 controls IL-31R� gene expression in mouse skin?

Minor points:

1) Images of “lesional-” panel in Fig.1A and B were adjusted brighter than other panels. All images should be taken under the same exposure condition. Figure 1 legend does not introduce what are blue channel and green channel representing.

2) Figure 1A and B, showed FK506 caused more than 40% reduction of PGP9.5+ and IL-31R� fibre density, why do the authors think this is not significant? (see page 11, lines 219-220 and 225-226)

3) Figure 1 legend: *p<0.01 should be *p<0.05

4) On page 18, this sentence is not clear. “PQA-18 significantly suppressed the development, while PQA-18 alone did not affect sensory nerve development.”

5) The authors need to clarify why DRG nerve fibre length in PQA-18 +rIL-31 treated DRG cells are shorter than PQA-18 alone and vehicle control in Fig.2 (see chart at bottom).

6) The authors need to explain why in Figure 4 PQA-18 inhibition of PAK2 and JAK2 phosphorylation is not dose dependent (over the conc used) however in Figure 5 PQA-18 inhibition of the neurite outgrowth is dose-dependent.

7) On page 9, line 169, addition of 40 pmol siRNA is in what volume of medium? It is maybe better to express as concentration (similar issue in Fig.6).

Reviewer #2: 

Ogura et al., (2016) showed that the prenylated quinolinecarboxylic acid derivative PQA-18  suppresses immune response, likely through inhibition of PAK2.  This was based on studies showing that there were changes in cofilin phosphorylation upon PQA-18  treatment (which could be via effects of PAK2 on LIMK1).  However the data presented then was not equivocal that  PQA-18  inhibits PAK2  directly, although other studies in KO mice to point to PAK1 and PAK2 regulating the immune system.

In this new study it is noted that PQA-18 improves the skin pruritus condition in a mouse model.  In order to demonstrate that PQA-18 works by inhibiting PAK2 (and likely PAK1) some additional experiment is needed.  In the neuroblastoma cell line etc.. it would be important to validate their 'PAK2 inhibitor' a well characterized ATP mimetic of  which FRAX are the best characterized.  It should be noted that IPA-3 does not work *in vivo* and should not be used.

The effects of PQA-18 on the IL-31/ IL-31R� pathway are quite convincing, and maybe clinically important.  I highlight below some issues that need to be considered.

(1) The treatment with PAK2 SiRNA shown in Figure 6 leads to profound changes in neurite outgrowth while the PAK2 KD by western is not that convincing (A).  To resolve whether the modest changes in PAK2 levels the authors need to use Frax 597/ 1036, which will strongly suppress the p-PAK Ser141 signal (which can be compared to PQA-18).

(2)  In Fig 3C  the authors show Stat3 western blots.  However since with no IL31 (lane 1)there is no observed pStat3 signal (bands) they need to present a different panel (ie one of the other blots which has been used to obtain average but not shown).  

(3)  Table 1. Data on the PAK2 interacting proteins should be more complete, and the 'top' MS derived set should be listed according to either enrichment relative to control or intensity / sequence coverage. 

(4)  The MS data which indicates that GIT2 and aPIX are present in the PAK complex is interesting (Fig 7). Based on current models, if PQA-18 inhibits PAK1/PAK2 directly one would expected that this would stabilize the PIX complex.  So this new data is interesting but should be supported with PAK2 IP & western data (for say aPIX).

(5)  In the raw data the p-JAK shows 2 strong bands which seem to be co-regulated.  What is the presumed identify of the top band?

(6)  In Figure 4 the anti-PAK2 data shows strong band at ~ 60 kDa with no other background bands.  By contrast the anti-PAK2 in Figure 6 shows several background bands - why is there such a large difference in the WB? Are different Abs used?

(7) The identity of the various antibodies used for analysis should be better defined in the figure legends (for example Fig 4) and ideally on the figures themselves.   cf.  The specifics of sites for phosphorylated sites in STAT3, JAK2, and PAK2. 

6. PLOS authors have the option to publish the peer review history of their article (what does this mean?). If published, this will include your full peer review and any attached files.

Reviewer #1: No

Reviewer #2: No

---

## [Author Response · Author response to Decision Letter 0]

17 Nov 2020

Responses to editor’s comments:

Thank you for your comments. As suggested, text and file naming have been revised. 

2. In your Methods section, please provide additional information on the animal research and ensure you have included details on : (1) methods of sacrifice (2) methods of anesthesia and/or analgesia, and (2) efforts to alleviate suffering.

Thank you for your comments. As suggested, the ‘methods’ section has been revised. In order to alleviate suffering, we used skilled cervical dislocation method. On the other hands, we did not use anesthesia to avoid its biological effects.

Thank you for your comments. As suggested, the Supporting Information has been revised. We have added our study’s minimal underlying data set to Supporting Information file.

Thank you for your comments. As suggested, the Supporting Information has been revised. We have added our all original full blot data set to Supporting Information file.

Thank you for your comments. As suggested, the ‘discussion’ section has been revised. We have removed the sentence (page 21, lane 411).

Thank you for your comments. As suggested, my Information has been revised.

Thank you for your comments. As suggested, captions for Supporting Information files have been added to the end of the manuscript.

Responses to reviewer’s comments:

Reviewer #1: Yoshimi Homma’s lab had previously reported that PQA-18, a derivative of Polysphondylium pseudo-candidum isolated small molecule Ppc-1, acts as an immunosuppressant and a PAK2 kinase inhibitor, could alleviate atopic dermatitis in mouse model (ref. 21). In the current study, the authors are trying to understand the mechanism of action of PQA-18 in relieving the skin defects in atopic dermatitis.

The authors chose to examine the IL-31/IL-31R� pathway and provide data to show that PQA-18 decreased the cutaneous sensory nerve fiber density by inhibiting the expression of IL-31 receptor alpha in mouse skin. In cultured neuronal cells, PQA-18 inhibits IL-31 induced sensory nerve fiber outgrowth and neurite outgrowth by blocking IL-31R� activation of the PAK2/JAK2/STAT3 pathway. Lastly, PQA-18 selectively blocks PAK2 interaction with GIT2 and �-PIX, which are thought to activate PAK2.

That PQA-18 acts as a potent immunosuppressant and effectively improves the skin pruritus condition in mouse AD model are potentially important findings. Experimental data for that PQA-18 inhibits IL-31/IL-31R� pathway and prevents cutaneous nerve fibre growing is convincing. But the molecular mechanism described for PQA-18 inhibiting PAK2 kinase is less convincing. There are a several issues which the authors need to address.

Major points.

1. The authors believed that PAK2 is an important target of PQA-18 and based on MS data in Fig.7 they suggest that PQA-18 inhibited PAK2 activation by preventing its interaction with GIT2/�-PIX. However more direct evidence is required to demonstrate PQA-18 blocks PAK2 interaction with α-PIX (GIT2 is likely indirect) using a direct binding assay.

Thank you for your comments. To incorporate this, we performed additional experiments using immunoprecipitation assay, and added findings to the ‘Results’ section (page 18, lane 374-377). Please refer new Fig 8B. We have changed previous labeled Fig 7 to new Fig 8.

2. Since both PAK1 and PAK2 are expressed in neuronal cells (PAK2 being ubiquitous) the authors should clarify if PQA-18 has similar effects on PAK1 kinase.

Thank you for your comments. As suggested, it is known that Neuro2A cells express both PAK1 and PAK2 (Ref 36). Although anti-phospho-PAK2 antibody recognizes both pSer141-PAK2 (61 kDa) and pSer144-PAK1 (68 kDa), we did not detect pSer144-PAK1 signal in our experimental condition (Figs 4 and 6). The pSer141-PAK2 signal was confirmed using anti-PAK2 antibody. Thus, it is conceivable that PAK2 rather than PAK1 is involved in the mechanism underlying the effect of PQA-18 on neurite outgrowth. Furthermore, we used Autodock Vina software (Le et al., Scientific reports, 2015) to analyze the PAK1 structure, and explore a binding pocket for PQA-18. However, we did not find suitable pocket. We will continue to analyze more details of the PAK-PQA interaction by considering dynamic structural changes, and report results on a separate paper.

3. In Fig.2 (compare IL-31R� images of vehicle vs PQA-18) the data does not support the claim that PQA-18 inhibits IL-31Ra expression in cutaneous nerve fibers. Cf. as stated on page 2 “PQA-18 alleviates cutaneous nerve fiber density and the expression of IL-31 receptor α (IL-31Rα) in the skin of Nc/Nga mice.” and page 11 “These results suggest that PQA-18 ointment, but not FK506 ointment, alleviates excessive cutaneous nerve fibre density and expression of IL-31Rα in the skin of Nc/Nga mice”.

Thank you for your comments. As suggested, we have revised these sentences and removed the description of IL31Rα expression (page 2, lane 17; page 11, lane 217 and lane 222; page 22, lane 419).

4. Do the authors think that PAK2 controls IL-31R� gene expression in mouse skin?

Thank you for your comments. As shown in Fig 2, IL31Rα was not only expressed in neuronal body but also nerve fiber structure in IL31-treated DRG neurons. Given the finding that PAK2 inhibitor PQA-18 suppresses IL31-induced nerve fiber development, we speculate that PAK2 may affect the expression of IL31Rα protein in sensory nerve fibers of lesional-skin. To confirm direct effects of PAK2 on IL31Rα gene expression in vivo, we are trying to produce conditional KO mice lacking PAK2, and would like to report their findings in the near future.

Minor points:

1) Images of “lesional-” panel in Fig.1A and B were adjusted brighter than other panels. All images should be taken under the same exposure condition. Figure 1 legend does not introduce what are blue channel and green channel representing.

Thank you for your comments. To incorporate this, Figs 1A, 1B and legend of Fig 1 have been revised (page 12, lane 228).

2) Figure 1A and B, showed FK506 caused more than 40% reduction of PGP9.5+ and IL-31R� fiber density, why do the authors think this is not significant? (see page 11, lines 219-220 and 225-226)

Thank you for your comments. As a result of the one-way analysis of variance with Turkey-Kramer post-hoc comparisons, treatment with FK506 did not significantly reduce PGP9.5-positive and IL31Rα-positive fiber density as compared with none and vehicle-treatment mice. In addition, vaseline, the base of the vehicle ointment and FK506 ointment, has been reported to slightly relieve atopic dermatitis symptoms through induction of protective factors including filaggrin (Nomura and Kabashima, J Allergy Clin Immunol,2016). In fact, previous (Ref. 21) and the results from this study also show that the vehicle group tends to improve atopic symptoms slightly, if not significantly. Thus, we consider it important to compare with the vehicle group in in vivo analysis.

3) Figure 1 legend: *p<0.01 should be *p<0.05

As suggested, the legend of Fig 1 has been revised (page 12, lane 231).

4) On page 18, this sentence is not clear. “PQA-18 significantly suppressed the development, while PQA-18 alone did not affect sensory nerve development.”

As suggested, this sentence has been revised (page 12, lane 240-241).

5) The authors need to clarify why DRG nerve fiber length in PQA-18 +rIL-31 treated DRG cells are shorter than PQA-18 alone and vehicle control in Fig.2 (see chart at bottom).

Thank you for your comments. As suggested, the length of nerve fiber in DRG cells treated with PQA-18 +rIL-31 was not significant, but it was shorter than that of the vehicle control and PQA-18 alone group. It is possible that negative feedback loop of JAK/STAT signaling is involved in its mechanism. The systems include induction of suppressor of cytokine signal (SOCS) and activation of tyrosine phosphatases (Jiang et al., Front Immunol. 2017). Thus, we speculate that both the PQA-18 and the negative feedback loop induced by IL31 may be shortened because it suppresses neurite outgrowth in cooperation. We will examine IL31 signaling including negative feedback loop and report the findings.

6) The authors need to explain why in Figure 4 PQA-18 inhibition of PAK2 and JAK2 phosphorylation is not dose dependent (over the conc used) however in Figure 5 PQA-18 inhibition of the neurite outgrowth is dose-dependent.

Thank you for your comments. As suggested, we have discussed the differences in the effects of PQA-18 and FRAX597 on PAK2 inhibition mechanisms and neurite outgrowth, and added the content to the ‘discussion’ section (page 24, lane 465-477).

7) On page 9, line 169, addition of 40 pmol siRNA is in what volume of medium? It is maybe better to express as concentration (similar issue in Fig.6).

As suggested, the ‘methods’ section and new Fig 7A have been revised (page 9, lane 163). We have changed previous labeled Fig 6 to new Fig 7.

Reviewer #2:

Ogura et al., (2016) showed that the prenylated quinolinecarboxylic acid derivative PQA-18 suppresses immune response, likely through inhibition of PAK2. This was based on studies showing that there were changes in cofilin phosphorylation upon PQA-18 treatment (which could be via effects of PAK2 on LIMK1). However the data presented then was not equivocal that PQA-18 inhibits PAK2 directly, although other studies in KO mice to point to PAK1 and PAK2 regulating the immune system.

In this new study it is noted that PQA-18 improves the skin pruritus condition in a mouse model. In order to demonstrate that PQA-18 works by inhibiting PAK2 (and likely PAK1) some additional experiment is needed. In the neuroblastoma cell line etc.. it would be important to validate their 'PAK2 inhibitor' a well characterized ATP mimetic of which FRAX are the best characterized. It should be noted that IPA-3 does not work in vivo and should not be used.

The effects of PQA-18 on the IL-31/ IL-31R� pathway are quite convincing, and maybe clinically important. I highlight below some issues that need to be considered.

(1) The treatment with PAK2 SiRNA shown in Figure 6 leads to profound changes in neurite outgrowth while the PAK2 KD by western is not that convincing (A). To resolve whether the modest changes in PAK2 levels the authors need to use Frax 597/1036, which will strongly suppress the p-PAK Ser141 signal (which can be compared to PQA-18).

Thank you for your comments. To incorporate this, we performed additional experiments using FRAX597, and added the findings to the ‘Results’ section (page 16, lane 317-321). Please refer new Figs 6A, 6B and 6C.

(2) In Fig 3C the authors show Stat3 western blots. However since with no IL31 (lane 1) there is no observed pStat3 signal (bands) they need to present a different panel (ie one of the other blots which has been used to obtain average but not shown).

Thank you for your comments. As suggested, Fig 3C has been revised.

(3) Table 1. Data on the PAK2 interacting proteins should be more complete, and the 'top' MS derived set should be listed according to either enrichment relative to control or intensity / sequence coverage.

Thank you for your comments. As suggested, Table 1 has been revised. Total coverage score and the number of detected unique peptides were added to Table 1.

(4) The MS data which indicates that GIT2 and αPIX are present in the PAK complex is interesting (Fig 7). Based on current models, if PQA-18 inhibits PAK1/PAK2 directly one would expected that this would stabilize the PIX complex. So this new data is interesting but should be supported with PAK2 IP & western data (for say aPIX).

Thank you for your comments. To incorporate this, we performed additional experiments using immunoprecipitation assay, and added the findings to the ‘Results’ section (page 18, lane 374-377). Please refer new Fig 8B. We have changed previous labeled Fig 7 to new Fig 8.

(5) In the raw data the p-JAK shows 2 strong bands which seem to be co-regulated. What is the presumed identify of the top band?

Thank you for your comments. We used anti-phospho-JAK2 (Tyr1008) antibody (CST) in this study. It is reported that cross-reactivity of this antibody is not observed with other JAK family members by immunoblotting. In addition, the top band (approximately 160 kDa) is not reacted with JAK2 antibody. Bands of 140 kDa and 120 kDa were observed with the JAK2 antibody (Fig 4 and S5 Fig full blot data), suggesting that it is not JAK family proteins. To identify the top band signal, we used amino acid sequence of JAK2 (DKVYYKV) and BLAST search. As a result, we selected macrophage-stimulating protein receptor (MST1R, RTK8, RON; 160 kDa) as candidates of top band. This protein expressed in DRG neurons is tyrosine kinase and involved in sensory nerve development (Franklin et al., Mol Cell Neurosci. 2009). We will clarify top bands using immunoprecipitation assay and LC/MS/MS.

(6) In Figure 4 the anti-PAK2 data shows strong band at ~ 60 kDa with no other background bands. By contrast the anti-PAK2 in Figure 6 shows several background bands - why is there such a large difference in the WB? Are different Abs used?

Thank you for your comments. As suggested, several background bands for PAK2 were detected in new Fig 7. However, when we performed experiments using PQA-18 and FRAX597, these background bands were not detected (Fig 4, new Fig 6A, S6 Fig and S7 Fig). Because we performed experiments at same antibody, it is possible that non-specific signals were induced by siRNA or electroporation method. We will try to examine other transfection reagents such as Lipofectaime2000 (Invitrogen) and Fugene HD (Promega) to avoid non-specific signal.

(7) The identity of the various antibodies used for analysis should be better defined in the figure legends (for example Fig 4) and ideally on the figures themselves. cf. The specifics of sites for phosphorylated sites in STAT3, JAK2, and PAK2.

Thank you for your comments. As suggested, figure legends have been revised. Please refer the legends of Figs 3 and 4, new Figs 6 and 7 (page 14, lane 271-272; page 14, lane 289-290; page 16, lane 325; page 17, lane 351).

---

## [Decision Letter · Decision Letter 1]

11 Jan 2021

PONE-D-20-23391R1

Prenylated quinolinecarboxylic acid compound-18 prevents sensory nerve fiber outgrowth through inhibition of the interleukin-31 pathway

PLOS ONE

Dear Dr. Ogura,

Thank you for resubmitting your updated  manuscript to PLOS ONE.  Apologies for the delays over the holiday period. The reviewers have indicated that they consider their comments have been addressed through the addition of new data, new figures, updated legends, and substantial correction of text.

There is one key issue outstanding regarding the new Figure 6 that will require the authors expanding/rewriting that section, namely that they have used Frax597 at a sub-optimal concentration.  The reason for doing this might be to avoid 'off-target' effects of Frax597 which are documented (to avoid affecting other kinases).

In Figure 6 I note you observe ~ 50% inhibition of PAK2 pS141 in their cells using 1 uM Frax597 with more substantive effect in cell assay. Thus the author should have considered 2 and 5 uM Frax597 doses. Please revise the MS to reflect the reasons why the 1 uM maximum (50% inhibition) was chosen for the neurite outgrowth assay (and p-PAK2 tested I think after for 30 min, rather than longer times, cf. 24/48h).

Admittedly the literature is a mess with respect to the proper concentration and timing to block PAK activity in cells. While the in vitro Ki is ~ 10-50 nM the effective dosing of cells with Frax579 usually requires 1-5 uM (ie 100 times more).

Licciulli et al., (2013) which the authors quote indeed indicates that pS141/4 signal was suppressed the cellular inhibition in SC4 cells in the range 0.5 -1 uM after 2h treatment. However this may reflect sensitivity of SC4 or a purer source of Frax579. In a 2015 paper (Oncotarget. Jul 10;6(19):16981-97) the authors use a 2h treatment of 2 uM Frax597 to effectively block both pS144 /141 signals.

Please submit a revised version of the manuscript that addresses the this point.

We look forward to receiving your revised manuscript.

Kind regards,

Ed

Edward Manser, PhD

Academic Editor

PLOS ONE

Reviewers' comments:

Reviewer's Responses to Questions

**Comments to the Author**

1. If the authors have adequately addressed your comments raised in a previous round of review and you feel that this manuscript is now acceptable for publication, you may indicate that here to bypass the “Comments to the Author” section, enter your conflict of interest statement in the “Confidential to Editor” section, and submit your "Accept" recommendation.

Reviewer #1: All comments have been addressed

2. Is the manuscript technically sound, and do the data support the conclusions?

Reviewer #1: Yes

3. Has the statistical analysis been performed appropriately and rigorously? 

Reviewer #1: Yes

4. Have the authors made all data underlying the findings in their manuscript fully available?

Reviewer #1: Yes

5. Is the manuscript presented in an intelligible fashion and written in standard English?

Reviewer #1: Yes

6. Review Comments to the Author

Reviewer #1: Authors have addressed most of the concerns I raised earlier.  In my opinion, provided the data on PAK inhibition in Figure 6 is better explained, the manuscript after minor revisions is therefore publishable.

7. PLOS authors have the option to publish the peer review history of their article (what does this mean?). If published, this will include your full peer review and any attached files.

Reviewer #1: No

---

## [Author Response · Author response to Decision Letter 1]

22 Jan 2021

Responses to editor’s comments:

1. There is one key issue outstanding regarding the new Figure 6 that will require the authors expanding/rewriting that section, namely that they have used Frax597 at a sub-optimal concentration. The reason for doing this might be to avoid 'off-target' effects of Frax597 which are documented (to avoid affecting other kinases).

In Figure 6 I note you observe ~ 50% inhibition of PAK2 pS141 in their cells using 1 uM Frax597 with more substantive effect in cell assay. Thus the author should have considered 2 and 5 uM Frax597 doses. Please revise the MS to reflect the reasons why the 1 uM maximum (50% inhibition) was chosen for the neurite outgrowth assay (and p-PAK2 tested I think after for 30 min, rather than longer times, cf. 24/48h).

Admittedly the literature is a mess with respect to the proper concentration and timing to block PAK activity in cells. While the in vitro Ki is ~ 10-50 nM the effective dosing of cells with Frax579 usually requires 1-5 uM (ie 100 times more).

Licciulli et al., (2013) which the authors quote indeed indicates that pS141/4 signal was suppressed the cellular inhibition in SC4 cells in the range 0.5 -1 uM after 2h treatment. However this may reflect sensitivity of SC4 or a purer source of Frax579. In a 2015 paper (Oncotarget. Jul 10;6(19):16981-97) the authors use a 2h treatment of 2 uM Frax597 to effectively block both pS144 /141 signals.

Please submit a revised version of the manuscript that addresses the this point.

Thank you for your comments. To incorporate this, we performed additional experiments using FRAX597 at 1, 2, and 5 μM, and added the findings of FRAX597 at 1 μM to the Figs 6A, 6B and 6C. The Supporting Information has been also revised.

As shown in Fig. R1, we observed that treatment with FRAX597 at 1, 2, and 5 μM (high concentration) for 30 min remarkably reduced phosphorylation of PAK2 in Neuro2A cells. However, we also observed that treatment with Frax597 at 2 and 5 μM for 48 h significantly reduced cell viability in Neuro2A cells (Fig. R2, Shakespear, Ogura et al., Neurochem Res., 2020). Thus, we did not add the findings of FRAX597 at 2 and 5 μM to the Fig 6. It has been reported that FRAX597 at high concentration can inhibit other kinases such as YES1 and RET (26). Given the reports that YES1 (Src family tyrosine kinase; Shani V et al., J Mol Neurosci., 2009) and RET (neurotrophin receptor tyrosine kinase; Tansey MG et al., Neuron, 2000) play an important role in neuronal survival, it is conceivable that FRAX597 at 2 and 5 μM may affect cell survival through inhibition of their tyrosine kinases in Neuro2A cells.

Responses to reviewer’s comments:

Reviewer #1:Authors have addressed most of the concerns I raised earlier. In my opinion, provided the data on PAK inhibition in Figure 6 is better explained, the manuscript after minor revisions is therefore publishable.

Thank you for your comments. As suggested, we performed additional experiments using FRAX597 at 1, 2, and 5 μM (Figs R1 and R2), and added the findings of FRAX597 at 1 μM to the Figs 6A, 6B and 6C. Since treatment with FRAX597 at 2 and 5 μM significantly reduced cell viability in Neuro2A cells, we did not add the results to the Fig 6.

---

## [Editor Report · Decision Letter 2]

25 Jan 2021

Prenylated quinolinecarboxylic acid compound-18 prevents sensory nerve fiber outgrowth through inhibition of the interleukin-31 pathway

PONE-D-20-23391R2

Dear Dr. Ogura,

Thank you for the changes presented in the new version of the MS.  I understand the reasoning for using the FRAX597 at lower conc. and the inhibition profile fits with published data in this cell line.   We’re pleased to inform you that your manuscript will be formally accepted for publication once it meets any outstanding technical requirements.

Within one week, you’ll receive an e-mail detailing any technical amendments. When these have been addressed, you’ll receive a formal acceptance letter and your manuscript will be scheduled for publication.

Kind regards,

Ed Manser, PhD

Academic Editor

PLOS ONE
---

## [Editor Report · Acceptance letter]

27 Jan 2021

PONE-D-20-23391R2 

Prenylated quinolinecarboxylic acid compound-18 prevents sensory nerve fiber outgrowth through inhibition of the interleukin-31 pathway 

Dear Dr. Ogura:

I'm pleased to inform you that your manuscript has been deemed suitable for publication in PLOS ONE. Congratulations! Your manuscript is now with our production department. 

Kind regards, 

on behalf of

Dr. Ed Manser 

Academic Editor

PLOS ONE